# Fungal mycelium classified in different material families based on glycerol treatment

Freek V. W. Appels[1], Jeroen G. van den Brandhof[1], Jan Dijksterhuis[2], Gijs W. de Kort[3] & Han A. B. Wösten [1 ✉]

Fungal mycelium is an emerging bio-based material. Here, mycelium films are produced from liquid shaken cultures that have a Young's modulus of 0.47 GPa, an ultimate tensile strength of 5.0 MPa and a strain at failure of 1.5%. Treating the mycelial films with 0–32% glycerol impacts the material properties. The largest effect is observed after treatment with 32% glycerol decreasing the Young's modulus and the ultimate tensile strength to 0.003 GPa and 1.8 MPa, respectively, whereas strain at failure increases to 29.6%. Moreover, glycerol treatment makes the surface of mycelium films hydrophilic and the hyphal matrix absorbing less water. Results show that mycelium films treated with 8% and 16–32% glycerol classify as polymer- and elastomer-like materials, respectively, while non-treated films and films treated with 1–4% glycerol classify as natural material. Thus, mycelium materials can cover a diversity of material families.

[1] Microbiology, Department of Biology, Utrecht University, Padualaan 8, 3584 CH Utrecht, The Netherlands. [2] Westerdijk Fungal Biodiversity Institute, Uppsalalaan 8, 3584 CT Utrecht, The Netherlands. [3] Aachen-Maastricht Institute of Biobased Materials (AMIBM), Faculty of Science and Engineering, Maastricht University, Brightlands Chemelot Campus, 6167 RD Geleen, The Netherlands. ✉email: h.a.b.wosten@uu.nl

The use of bio-based materials is part of the transition towards a sustainable economy. Molecules or structures of microbes, algae, plants and animals are a source for these materials. Thermoplastic starch from plants[1], polyhydroxyalkanoate (PHA) from bacteria[2] and fungal mycelium[3–8] are examples of bio-based materials. Fungi have evolved as one of the most effective waste degraders in nature. This trait can be used to convert low-quality agricultural waste streams into bio-based materials.

Pure and composite mycelium materials are distinguished as well as materials derived from polymers of mycelium. Chitin-glucan complexes from fungal mycelium have attracted attention in the bio-based material field. Paper-like nanomaterials consisting of this complex are obtained by soft-compression of homogenised mycelium or fruiting bodies that have been extracted with hot water and mild alkaline[9,10]. The chitin-glucan materials from *Agaricus bisporus* mushrooms are characterised by Young's modulus (*E*) and ultimate tensile strength (σ) of approximately 7 GPa and 100–200 MPa, respectively[9]. These mushroom derived materials are the most rigid and strongest fungal derived material to date.

Properties of pure, non-extracted fungal materials depend on the substrate, the type of fungus, and its growth conditions[5,6] as well as post-processing[3]. For instance, Young's moduli of mycelium films of static liquid cultures of the edible mushroom forming fungus *Schizophyllum commune* ranges between 0.44 and 0.91 GPa when grown at 400 or 70.000 ppm $CO_2$[3]. Notably, this range is between 1.24 and 2.73 GPa when the hydrophobin gene *sc3* is inactivated. This enhanced rigidity is caused by the increased density of the mycelium of the *sc3* deletion strain, shifting the mechanical properties being similar to natural materials (e.g. leather) in the case of the wild-type to those of polymers (e.g. high density polyethylene) in the case of the deletion strain.

Plasticizing agents reduce brittleness of films from bio-derived polysaccharides. Plasticizers are low molecular weight non-volatile compounds that are incorporated in a material to increase flexibility. Glycerol is amongst the most commonly used plasticizers[11,12] and is a byproduct of bio-diesel production[13]. This small polyol has a hygroscopic nature and increases biofilm flexibility by increasing space between its polymers[14]. For instance, glycerol increases elasticity, reduces ultimate tensile strength and increases strain at failure of crustacean chitin derived nanopapers[15].

Here, liquid cultures were shaken at high speed to resemble growth in bio-reactors. Biomass was harvested (leaving mycelial pellets intact), dried and either or not treated with glycerol. Different from Nawawi et al.[9] and Jones et al.[10], no extraction steps were applied. Glycerol treatment changed the material properties from being similar to natural materials to those of polymers and elastomers. Moreover, the treatment impacted hydrophilicity, water absorption and thickness expansion after water submersion.

## Results

**Biomass production of S. commune.** Liquid shaken cultures of *S. commune* 4–39 were grown for 7 days in the dark with 200 mg wet weight mycelial homogenate as inoculum. The mycelium that was harvested by vacuum filtration consisted of 6.0% ± 0.9% dry weight (*n* = 10, ±s.d.). Total dry weight biomass of cultures was 8.5 ± 0.7 g L$^{-1}$ (*n* = 10, ±s.d.), implying a 37% conversion of the dry matter in the medium into mycelium.

**Post-treatment of mycelium films of S. commune.** Mycelium films were submerged for 24 h in 0–32% glycerol and dried at

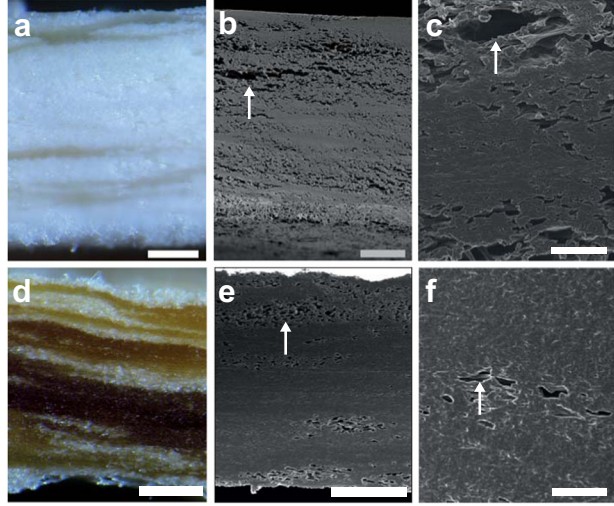

**Fig. 1 Microscopy of mycelial films of S. commune.** Light microscopy (**a**, **d**) and scanning electron microscopy (**b**, **c**, **e**, **f**) of longitudinal sections of untreated mycelium films (**a–c**) and films treated with 8% glycerol (**d–f**). Mycelium was grown as liquid shaken cultures, harvested by vacuum filtration, and dried at room temperature. The resulting mycelium films were either or not treated with 8% glycerol and dried at room temperature. Untreated mycelium has a white appearance (**a**) turning brown after treatment with 8% glycerol (**d**). In addition, treatment with 8% glycerol results in less air voids (indicated with white arrows) being trapped in the mycelium material. Scale bars represent 100 μm (**a**, **b**, **d**, **e**) and 10 μm (**c**, **f**).

room temperature. In general, hyphae were difficult to distinguish in the material as they merged into a compact material that had trapped air voids (Fig. 1). Untreated material had a white and brittle appearance. This changed to a brownish, rubbery appearance when mycelium had been treated with ≥8% glycerol (Fig. 1d). The number of air voids decreased after treatment with 8% glycerol when compared to untreated mycelium (Fig. 1).

**Material properties of glycerol treated mycelial films.** The density of dried mycelium films (587 ± 27 kg m$^{-3}$) was not statistically different after treatment with water or 1% glycerol (Table 1). Density did increase after treatment with ≥2% glycerol. For instance, densities of 1338 ± 38 kg m$^{-3}$ and 1435 ± 29 kg m$^{-3}$ were observed after treatment with 16 and 32% glycerol, respectively (Table 1). This increased density was not the result of a decreased thickness of the mycelium as was observed in the case of treatment with 2–8% glycerol.

Stress/strain curves of mycelium materials showed that untreated mycelium material and films treated with 0–4% glycerol behaves as brittle materials, whereas treatment with 16 and 32% glycerol resulted in ductile behaviour (Fig. 2). Mycelium treated with 8% glycerol was less brittle than mycelium treated with lower concentrations of glycerol but less ductile than materials treated with higher concentrations of glycerol.

Submersion of mycelial films in 0–2% glycerol resulted in a higher *E* (0.95 ± 0.08 GPa–1.26 ± 0.15 GPa) when compared to untreated mycelium (0.47 ± 0.04 GPa), while treatment with 8–32% glycerol resulted in a lower *E* modulus ranging between 0.12 ± 0.02 and 0.003 ± 0.000 GPa (Table 1, Supplementary Fig. 1). The ultimate tensile strength (σ) of mycelium treated with 0–4% glycerol ranged between 9.6 and 12.3 MPa and was in all cases higher than that of untreated mycelium (5.0 ± 0.5 MPa) (Table 1, Supplementary Fig. 1). In contrast, the ultimate tensile strength of mycelium treated with 8 and 16% glycerol (6.4 ± 0.7 and 3.6 ± 0.3 MPa, respectively) was not different from the strength of

**Table 1 Material properties of mycelial films of *S. commune*.**

| | Thickness (mm) | Weight (g) | Density (kg m⁻³) | E (GPa) | σ (MPa) | ε (%) | n | WCA (°) |
|---|---|---|---|---|---|---|---|---|
| Control (a) | 0.59 ± 0.03^b-g | 0.54 ± 0.01^b-d,g,h | 587 ± 27^d-h | 0.468 ± 0.043^b-d,f-h | 5.0 ± 0.5^b-e,h | 1.5 ± 0.1^f-h | 12 | 129 ± 2^b-h |
| H₂O (b) | 0.36 ± 0.03^a,h | 0.38 ± 0.01^a,c,e-h | 696 ± 48^e-h | 1.257 ± 0.146^a,e-h | 11.2 ± 1.4^a,g,h | 1.2 ± 0.1^e-h | 16 | 99 ± 8^a,g,h |
| 1% glyc (c) | 0.37 ± 0.02^a,h | 0.44 ± 0.01^a,b,e-h | 769 ± 46^e-h | 0.949 ± 0.083^a,f-h | 9.6 ± 0.8^a,g,h | 1.4 ± 0.1^e-h | 12 | 102 ± 2^a,g,h |
| 2% glyc (d) | 0.30 ± 0.03^a,h | 0.45 ± 0.02^a,f-h | 994 ± 76^a,g,h | 1.048 ± 0.069^a,e-h | 12.3 ± 1.2^a,f-h | 2.2 ± 0.3^a,g,h | 8 | 101 ± 2^a,g,h |
| 4% glyc (e) | 0.29 ± 0.02^a,h | 0.52 ± 0.02^b,c,g,h | 1166 ± 74^a-c | 0.688 ± 0.059^b,d-h | 10.1 ± 1.0^a,g,h | 3.8 ± 0.6^a-c | 8 | 93 ± 7^a,h |
| 8% glyc (f) | 0.29 ± 0.01^a,g,h | 0.58 ± 0.02^b-d,g,h | 1262 ± 37^a-c,h | 0.124 ± 0.020^a-h | 6.4 ± 0.7^d,g,h | 14.9 ± 1.7^a-c,h | 12 | 86 ± 5^a,h |
| 16% glyc (g) | 0.38 ± 0.02^a,f,h | 0.81 ± 0.03^a-h | 1338 ± 38^a-d | 0.021 ± 0.004^a-h | 3.6 ± 0.3^b-d | 23.7 ± 1.7^a-d | 8 | 71 ± 6^a-d |
| 32% glyc (h) | 0.51 ± 0.02^b-g | 1.16 ± 0.04^a-h | 1435 ± 29^a-d,f | 0.003 ± 0.000^a-g | 1.8 ± 0.1^a-g | 29.6 ± 0.9^a-d,f | 8 | 49 ± 3^a-f |

Thickness (mm), weight (g), density (kg m⁻³), Young's modulus (E, GPa), ultimate tensile strength (σ, MPa), strain at failure (ε, %) and static water contact angle (WCA) of mycelial films either or not treated with 0–32% glycerol (mean ± SEM). The number of biological replicates is shown (n). Letters following data indicate statistically significant differences to the materials specified with that letter (p ≤ 0.05). Density of the materials increases when treated with ≥2% glycerol. Untreated mycelium material and films treated with 0–4% glycerol behave as brittle materials, whereas treatment with 16 and 32% glycerol results in ductile behaviour. Treatment with 0–2% glycerol results in a higher E when compared to untreated mycelium, while treatment with 8–32% glycerol results in a lower E modulus. The ultimate tensile strength (σ) of mycelium treated with 0–4% glycerol is in all cases higher than that of untreated mycelium. In contrast, mycelium treated with 32% glycerol result in the weakest material. Untreated mycelium material is hydrophobic, while treatment with ≥8% glycerol makes the mycelium hydrophilic. Mycelium films treated with 32% glycerol take up the lowest amount of water, while highest weight increase is observed for mycelium treated 0–2% glycerol.

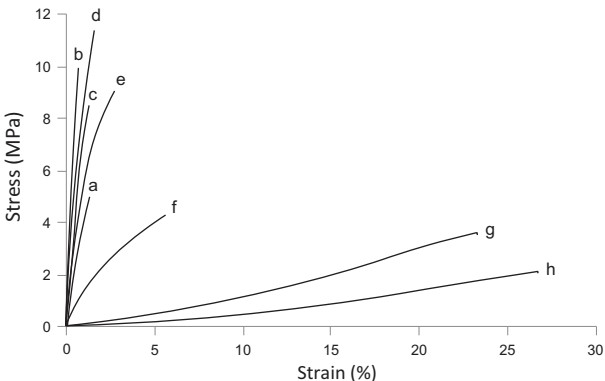

**Fig. 2 Stress/strain curves of mycelial films of *S. commune* either or not treated with glycerol.** Representative stress/strain curves of non-treated mycelium material (**a**) and mycelium material treated with H₂O (**b**) and 1% (**c**), 2% (**d**), 4% (**e**), 8% (**f**), 16% (**g**) and 32% (**h**) glycerol. Mycelium films treated with 0–4% glycerol show brittle behaviour with ultimate tensile strengths ranging between 9.6 MPa (**c**) and 12.3 MPa (**d**) and strain at failure ranging between 1.2% (**b**) and 3.8% (**e**). Treatment with 16 and 32% glycerol results in ductile materials with strain at failure of 23.7 and 29.6%. Treatment with 8% glycerol results in a material exhibiting less brittle behaviour than treatment with 0–4% glycerol but lower ductility than mycelium films treated with 16 and 32% glycerol. Young's moduli are derived from the linear part of these stress/strain curves (Table 1).

untreated mycelium. Treatment with 32% glycerol resulted in the weakest material showing an ultimate tensile strength of 1.8 ± 0.1 MPa. Strain at failure did not differ between untreated mycelium films and films treated with 0–1% glycerol with values ranging between 1.2 and 1.5 (Table 1). Higher percentages of glycerol showed increased strain at failure up to 29.6 ± 0.9% at 32% glycerol.

**Water contact angle and water absorption of mycelial films**. Untreated mycelium material was hydrophobic as shown by its static water contact angle (WCA) of 129 ± 2° (Table 1). Glycerol treatment reduced the WCA showing hydrophilicity of mycelium (water contact angle < 90°) after treatment with ≥8% of the plasticizing agent. Treating mycelium with 32% glycerol resulted in the lowest WCA of 49 ± 3°. Untreated mycelium and mycelium treated with water or low concentrations of glycerol slowly absorbed the water droplets used for the WCA measurements, while treatment with higher concentrations of glycerol did not show water absorption of the mycelium during the measurements. Water absorption was assessed quantitatively by submersion of the mycelium films in water over a period of 24 h. In all cases, most weight increase was observed during the first 10 min of water submersion (Fig. 3). Mycelium films treated with 32% glycerol took up the lowest amount of water (100 ± 5 wt%), while highest weight increase was observed for H₂O treated mycelium (487 ± 33 wt%) and 1% (458 ± 11 wt%) and 2% (449 ± 46 wt%) glycerol treated mycelium. Thickness expansion after 24 h of water submersion was lowest for untreated mycelium (102 ± 8%) and mycelium treated with 32% (189 ± 19%) and 16% (277 ± 18%) glycerol, while high expansion was observed for mycelium treated with H₂0 (493 ± 33%) and 1% (496 ± 16%) and 2% (472 ± 55%) glycerol (Supplementary Table 1).

## Discussion

Mycelium of the mushroom forming fungus *S. commune* was grown in liquid shaken cultures, dried in films and either or not post-treated with the plasticizing agent glycerol. Properties of the films that were not treated with glycerol were similar to those of

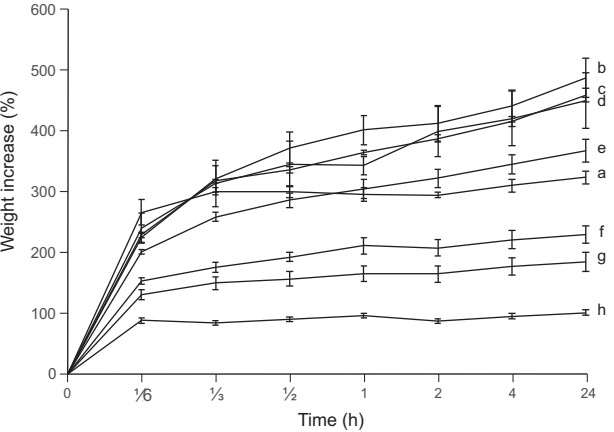

**Fig. 3 Water submersion of mycelial films of *S. commune*.** Weight increase (wt%) after water submersion of non-treated mycelium material (**a**) and mycelium material treated with $H_2O$ (**b**) and 1% (**c**), 2% (**d**), 4% (**e**), 8% (**f**), 16% (**g**) and 32% (**h**) glycerol. Error bars represent SEM ($n = 4$). Mycelium films were submerged in water and weighed after 10, 20 and 30 min and 1, 2, 4 and 24 h. Most water is absorbed during the first 10 min of water submersion. Mycelium films treated with $H_2O$ (487 wt%, **b**) and 1% (458 wt%, **c**) and 2% (449 wt%, **d**) glycerol take up the highest amount of water, while the lowest amount of water is absorbed by material treated with 32% glycerol (100wt%, **h**).

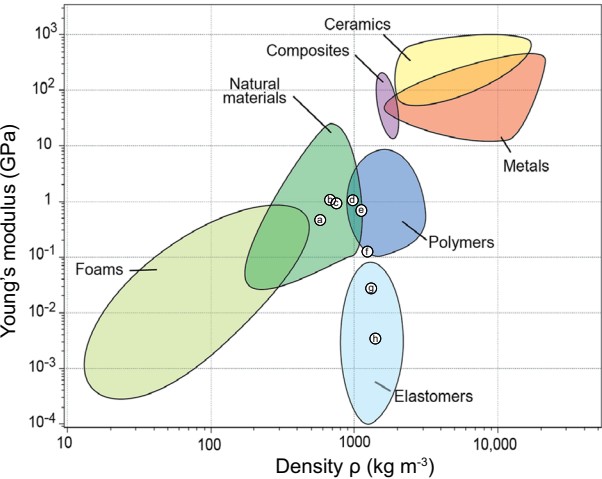

**Fig. 4 Young's moduli and density of different material families.** Young's modulus (*E*) (GPa) vs density (kg m$^{-3}$) of non-treated mycelium material (**a**) and mycelium material treated with $H_2O$ (**b**) and 1% (**c**), 2% (**d**), 4% (**e**), 8% (**f**), 16% (**g**) and 32% (**h**) glycerol projected in the material family chart. Treatment with increasing concentrations of glycerol resulted in mycelium films with a Young's modulus and density shifting from being similar to natural materials (**a**–**d**), to polymers (**d**–**f**) to elastomers (**g**, **h**). Figure reproduced with permission[18].

static liquid cultures of *S. commune*[3]. The liquid shaken cultures result in more reproducible materials and are easier to upscale and are therefore preferred over static liquid cultures. Treatment of the mycelium films with glycerol impacted Young's modulus and material density, thereby classifying the films as natural materials and polymer- and elastomer-like materials, depending on the glycerol concentration used. To the best of our knowledge, this is the first time mycelium materials have been obtained with elasticity and density similar to elastomer-like materials (Fig. 4).

Mycelium films treated with 0–2% glycerol were thinner and had a reduced weight compared to untreated mycelium. The latter can be explained by washing water-soluble proteins and sugars out of the material during the 24 h of submersion in the glycerol solution. The same may occur at higher concentrations of glycerol but here the amount of glycerol penetrating the mycelium compensates for (4–8% glycerol) or is higher (16–32% glycerol) than the removal of water-soluble polymers from the mycelium. Glycerol forms hydrogen bonds with cell wall polymers, thereby increasing the intramolecular space between these cell wall components[14]. The uptake of glycerol and its interaction with cell wall molecules could also explain why mycelium treated with increasing concentrations of the plasticizer absorbed less water and showed a reduction in thickness expansion after submersion in water. Despite the highly hydrophobic nature of its surface[16], untreated mycelium took up relatively high amounts of water. Initially, the static water contact angle was 129 ± 2°, indicative of a very hydrophobic material. However, after some time the water got absorbed most likely due to capillary forces resulting from the open hyphal architecture corresponding with its relatively low density. Treatment with glycerol reduced the water contact angle; a 32% solution reduced the water contact angle even to 49 ± 3°. This may be explained by the reduction of air voids lowering these capillary forces and an increase in the presence of glycerol that is hydrophilic by nature. Also, it was observed that glycerol treatment reduced surface roughness, thus possibly lowering surface hydrophobicity.

Treatment with glycerol resulted in materials with different mechanical properties. Treatment with 0–2% glycerol resulted in

a 2-fold increased *E* and σ, and a 1.2–1.7-fold increased density when compared to untreated mycelium. In addition, elasticity increased by 50% in the case of 2% glycerol. Treatment with 8% glycerol resulted in a 2.1-fold increase in density, a 3.8-fold decrease in *E*, a 1.3-fold increase in σ, and an almost 10-fold increase in elasticity. Finally, treatment with 32% glycerol even showed a 2.4-fold increase in density, a 150-fold lower *E*, a 2.8-fold lower σ, and a 20-fold higher elasticity when compared to untreated mycelium. The low *E* together with relatively high density classifies mycelium material treated with 16 or 32% glycerol in the family of elastomers. This was also deduced from the stress/strain curves indicating a ductile material. A decreased E while observing increased density in the case of treatment with ≥8% glycerol contrasts the positive correlation between density and Young's modulus in the case of untreated mycelium[3].

Together, mycelium materials with different density, hydrophilicity and mechanical properties can be produced by treating pure mycelial films with glycerol. Increasing the concentration of glycerol changed material properties from paper-like, to leather-like, to rubber-like, illustrating the wide potential of mycelium material applications.

## Methods
**Strains and culture conditions**. *S. commune* wild-type strain 4–39 (CBS 341.81) was grown for 5 days at 30 °C in the light (1000 lx) in 55 mm diameter Petri dishes containing 10 mL minimal medium (MM) with glucose as carbon source and asparagine as nitrogen source[17] and solidified with 1.5% agar. The culture was homogenised in 100 mL MM for 30 s at 18,000 rpm using a Waring Blender (Waring Laboratory, Torrington, England) and grown in a 250 mL Erlenmeyer for 24 h at 200 rpm. This pre-culture was homogenised for 30 s at 18,000 rpm and aliquots of mycelium (200 mg wet weight) were used to inoculate a 2 L Erlenmeyer containing 1.2 L MM. Cultures were grown for 7 days at 30 °C in the dark at 200 rpm. After adding 800 mL of water, the culture was filtered using a Melitta® coffee filter placed in a Büchner funnel (110 mm diameter) that was connected to a vacuum pump (Leybold, Divac 1.2L, Cologne, Germany). The resulting layer of mycelium was transferred to a flat surface covered with cellophane (Embalru, Nijverdal, Netherlands) and dried at room temperature at ±50% relative humidity.

**Post-treatment of mycelium materials**. Strips of dried mycelium (2 × 8 cm) were submerged for 24 h in 0–32% aqueous glycerol (Sigma, St Louis, MO, USA) and dried between two films cellophane at ambient temperature.

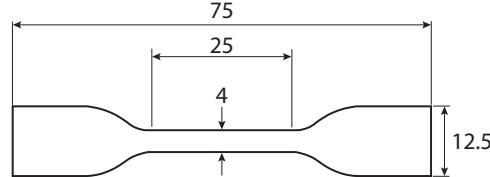

**Fig. 5 Size of mycelium specimens used in tensile tests.** Dog bone shape specimen with dimensions (mm) according to ISO 527 type 5A. Mycelium films with this shape were used in tensile tests. To this end, Specimens were fixed at the wide ends between clamps of a tensile test machine. These clamps were pulled apart until failure of the specimen, while recording the stress and strain. Specimens fail at the middle part of the sample as this is the weakest part, assuring clamping does not impact the recorded material properties.

**Mechanical analysis of mycelium materials**. Dog bone shaped mycelial films of 8–16 biological replicates (see Table 1) were produced using an ISO 527 type 5A (Fig. 5) sample cutter attached to a Zwick ZCP 020 manual cutting press (Zwick GmbH, Ulm, Germany). Thickness of the resulting bone shaped samples was measured at three points along the axis of the sample using a digital length gauge device (Heidenhain-Metro MT 1200, Traunreut, Germany). Sample weight was measured after tensile tests that were performed at room temperature with a Zwick/Roell Z020 (Zwick GmbH, Ulm, Germany) using a preload force of 0.25 N with a 2 mm min$^{-1}$ test speed. Specimens were fixed by clamps 10 mm from each end. The bulk density was calculated by dividing the weight of the specimen by the bulk volume of the sample. Young's modulus ($E$) (in GPa) was determined at the linear part of the stress/strain curve. The ultimate tensile strength ($\sigma$) (in MPa) was obtained from the maximum load (N) per unit area (mm$^2$) of the specimen, while strain at failure ($\varepsilon$) (%) was obtained by calculating the strain (mm) at the moment of breaking.

**Water contact angle of mycelial films**. Static water contact angles (WCA) were measured at 23 °C with a Drop Shape Analyser DSA 10 Mk2 (KRÜSS, Hamburg, Germany). A baseline was set manually before measuring WCA of 15 droplets of 5 µL ultra-pure water for each of the 4 biological replicates. The WCA was measured 5 s after placing the droplet on the mycelium film.

**Water absorption of mycelial films**. Water absorption of mycelial films was determined according to ASTM D570. To this end, four biological replicates of films (2 × 2 cm) of each treatment were submerged in demineralised water at 23 °C. Water absorption was measured at different time points during a 24 h period by determining the percentage of weight increase. Excess water was removed with filter paper before weighing.

**Statistics and reproducibility**. Statistical analysis was performed with the software package IBM SPSS statistics 22.0 (IBM Corporation, Armonk, New York). Material and mechanical properties of mycelium specimens were analysed by a Welch's unequal variances t-test followed by a Games–Howell post hoc test ($p \leq 0.05$). Statistical analysis on the static water contact angle was performed with a One-Way Anova followed by a Tukey HSD post hoc test. To this end, means were used that were calculated from 15 measurements for each of the four biological replicates. Normal distribution was tested using a Kolmogorov–Smirnov test and homogeneity of variance was tested using Levene's test.

**Reporting summary**. Further information on research design is available in the Nature Research Reporting Summary linked to this article.

## Data availability
The datasets generated and analysed during the current study can be found in Supplementary Data. Remaining information is available from the corresponding author upon reasonable request.

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

## Author contributions
F.V.W.A., J.G.V.D.B., J.D. and G.W.D.K. performed the experiments. F.V.W.A., J.G.V.D.B. and H.A.B.W. designed the experiments, interpreted results, and wrote the manuscript.

## Competing interests
The authors declare no competing interests.
