## [Peer Review File · Communications Biology]

Reviewers' comments:

Reviewer #1 (Remarks to the Author):

I would suggest that it is a cool concept, customisable mechanical properties in the natural material, elastomer and polymer ranges. The paper should be accepted after minor revisions. The following must be addressed:

You have measured density, however there is no information regarding how you measured it in the materials and methods. This must be included. Additionally, did you measure envelope or skeletal density? This must be specified. It would be a good idea to measure both envelope and skeletal densities and then calculate the porosity of your material.

Figure 5. is definitely not adapted. It is reproduced. Adapted would indicate that you have changed it significantly. Adding your data to it does not suffice as an adaptation. You will need to ensure that you have copyright permission for the use of the image.

The water contact angle can be prone to very significant error. Have you used a static or dynamic contact angle measurement technique? A static water contact angle may be subject to significant contact angle hysteresis and will likely be inaccurate. Please provide advancing or receding dynamic contact angle measurements. Preferably both. 3 replicate drop profiles is also insufficient for contact angle measurements. Please provide at least 10 replicates.

The contact angle measurements do show changes in hydrophilicity. However, water permeability is indicated by water flux. I do not see any water flux measurements in this article. These must be included in order to claim differences in water permeability.

You say: "The intra- and extracellular glycerol would also explain why mycelium treated with 8–32% glycerol did not sorb water, while sorption was observed in the case of untreated mycelium despite its highly hydrophobic nature of its surface". However, you have not measured sorption at all. Contact angle measurements only provide you with the contact angle and if you use more than one test fluid then an approximation of the surface energy. You cannot talk about sorption when you haven't measured sorption.

You say: "Scanning electron microscopy indicates that glycerol fills the air voids that are present in untreated mycelium". These SEM micrographs should be included in the article or supplementary material.

You say: "For instance, the reduced water sorption of mycelium materials after treatment with glycerol is of interest for outdoor applications." However, this is contradictory to your results. The more glycerol you add the more hydrophilic your materials become (reducing contact angle). So, treatment with glycerol certainly isn't in the interest of outdoor applications. Also, if you are going to talk about water sorption then it should be accompanied by an isotherm. From what I can see you haven't measured anything at all related to water sorption.

This article addresses the mechanical and surface properties of treated mycelium biomass. You should compare your results to other studies examining the mechanical and surface properties of treated mycelium biomass. e.g.

Jones, M., Weiland, K., Kujundzic, M., Theiner, J., Kählig, H., Kontturi, E., John, S., Bismarck, A. and Mautner, A., 2019. Waste-derived low-cost mycelium nanopapers with tunable mechanical and surface properties. *Biomacromolecules*, 20(9), pp.3513-3523.

The reference Jones, Huynh, Dekiwadia, Daver, & John, 2017 appears in text but does not appear in the reference list.

Reviewer #2 (Remarks to the Author):

This manuscript reports a sustainable fungal mycelium, the production process and experimental measure methods were present, and the test results were analyzed and discussed. Finally, the produced fungal mycelium was figured on the material family chart, and shown the mechanics behaviors compared with other materials. This work is interesting for the sustainable green materials, and I think it could meet our journal. However, this manuscript should clear some mechanical test process and method in detail.

The comments please find as follows:

- 1) Line 87-88: How much the light intensity?
- 2) Line 102: What size of the samples for mechanical tests?
- 3) Line 105: During the preparation of tensile test, please clear that how to fix the top and bottom of the size?
- 4) Line 106: What is the temperature during the test?
- 5) Line 107 What experiment standard was used to test the Young's modulus? Generally, Young's modulus could be determined from the compressive test.
- 6) Line 108-109: the symbol of ultimate tensile strength should be the σ , and the strain should be ϵ .
- 7) Line 101-109: How many samples were measured for each group test.
- 8) Line 108: How to determine the ultimate tensile strength from the stress-strain curves (For the figure 3)? Generally, the peak of stress-strain curve or the stress at certain strain is determined as the tensile strength. So, please clear the determine method and test standard in detail.

9) Line 146 Figure 2: Please show SEM photos to present the micro-structure of the mycelium.
10) Line 180 Figure 4C, Please replace the vertical coordination axis as 'broken strain'.
Finally, I encourage the authors to show the thermal conductivity and toxicity of these fungal mycelium.

Reviewer #3 (Remarks to the Author):

The study showcases that a treatment with glycerol impact mycelium-material properties, resulting in sheets with stiffer and more elastic properties, similar to industrial polymers and elastomers. This study shifts mycelium materials to propose relevant alternatives to synthetic raw materials, an important milestone in promoting this research field.

The methods and results are properly detailed and clearly described. The obtained results provide useful information to develop further applications and production methods.

The use of pure mycelium material originated in liquid culture instead of producing plant-based mycelium composites is novel and promising, yet to my knowledge it should not be targeted as the novelty of this article. I suggest adding some references to other publications that did similar studies using mycelium material originated in liquid culture (detailed below) or refer to other bio-based materials that offer similar properties – it will contribute to emphasize the uniqueness of your approach and locate your results within a focused context.

1. Lines 21-22

- This sentence could be refined since mycelium materials in academia and industry are regularly produced in liquid cultures. Please consider referring to Livne, A. et al. (2019) – To my knowledge they demonstrated the use of shaken liquid culture in a recent publication [A fungal mycelium templates the growth of aragonite needles. J. Mater. Chem. B 7, 5725–5731]
- Maybe replace 'material' with 'films'

2. Lines 21-22

- Try to avoid clustered references, maybe separate the mycelium-based references from the other sourced materials
- What are the expected advantages of mycelium-based materials over the other bio-based materials that you mention?

3. Lines 50-66

- I don't think that this paragraph and the following one are essential – It would be better to give some background on the use of glycerol as a plasticizing agent, why glycerol, its effect on physical and mechanical properties, is it biodegradable? What other relevant properties does it offer, maybe a short overview on previous starch/chitosan/glycerol films studies
- What are the relevant differences between using chitin, chitosan and mycelium?

4. Line 57, line 69

- Here again I believe it would be clearer to separate the references, for example to pure and composite materials.

5. Line 78

- Worth regarding to W. Nawawi et al. 2019. that recently produced mycelium materials originated in liquid culture, I think it is comparable and relevant, and maybe suggest what considerations led you to use mycelium rather than extracted chitin?

6. Line 87

- Do you think that the Minimal Medium might have contributed to the increase in strength/or density?

7. Line 88

- "The culture was homogenized in 100 mL MM for 30 sec" At what rate?

8. Line 90

- What does the minimal medium contain in this stage? Is it in a liquid state here? Does this substrate contain glucose? What else?

9. Line 91

- "grown for 7 days in the dark at 200 rpm" At what temperature?

10. Line 93

- Was the mycelium deactivated?
- Add drying conditions (temperature, humidity, clamping?)

11. Line 135 (figure 1)

- Is it a light microscope or camera?
- Do you discuss the effect of inoculum volume on pellet dimensions in the text? Are there other factors that might affect it? Why is the pellet dimension important? Does it affect other properties or affected by certain factors during growth?

12. Line 139

- Replace 'incubated' with 'submerged'?

13. Line 145 (figure 2)

- The scale bars are different thicknesses and should be fixed
- Maybe add a top view photo to better understand what your material looks like

14. Line 155 (table 2)

- mm units would be more appropriate for film thickness
- I wonder why all the standard deviations for thickness are zero. Did you take only 1 measurement? you can compare your measurements to the microscope images to add data and validate your measurements.

- This data is also presented in figure 3 and 4. Maybe the table could go to supplemental data or that the figures should present something beyond the data of the table.

15. Line 167

- Colors should be more varied – maybe add shapes or letters near each curve?

16. Line 170

Replace 'incubation' with 'treatment' or 'submerged'?

17. Line 180 (figure 4)

- This information already appears in table 1 - Can you show new data here? For example, curves that represent interesting crossing of different properties.

18. Line 189

- Contact angle is affected by the surface roughness- do you know whether the roughness of the films was affected by the different compositions you tested?

- And glycerol is hydrophilic but maybe it also contributed to surface smoothness thus decreased its hydrophobicity?

19. Line 191

- Not clear if the water were taken-in faster or slower with glycerol. I think you should refine if you are saying that the water droplets did not penetrate the films after 5 min at 32% glycerol.

20. Line 195-208

- Most of the paragraph discusses the previous study. I would recommend to significantly reduce it or focus on why it is relevant in the discussion

21. Line 205

- Then what is the difference between static and liquid culture? The yield? Based on your studies- which one suits better for industrial production of mycelium materials?

22. Line 207

- Polymer-like...

23. Line 208

- "Thus, we have produced for the first-time mycelium materials that are classified in the latter material group" > I think that this is the important point to emphasize for 'first time' and try to avoid erosion of this phrase over this article.

24. Line 195-208

- "16 or 32% glycerol is higher than that of the plasticizing agent" > Confusing- If I got it right, the glycerol is the plasticizing agent. Please refine if you are saying that the density of the mycelium-glycerol is greater than pure glycerol?

25. Line 227

- I agree that water is probably trapped between the hyphae, but what do you mean with the phrase 'hydrophobic air'? Did you mean just air?

Point-by-point response to the referees' comments

First of all, we would like to thank the reviewers for helping us to improve the quality of the manuscript. We have addressed all their points and performed additional experiments. Our response is in italics and we refer to line numbers (abbreviated with l.) in the revised manuscript.

General

To comply to the format requirements of Communications Biology we have adapted the manuscript, for instance by reducing the number of words of the title to a maximum of 15 words, reducing the number of words in the abstract to a maximum of 150 words and by changing the order of the sections.

Reviewer #1

I would suggest that it is a cool concept, customisable mechanical properties in the natural material, elastomer and polymer ranges. The paper should be accepted after minor revisions. The following must be addressed:

1) You have measured density, however there is no information regarding how you measured it in the materials and methods. This must be included. Additionally, did you measure envelope or skeletal density? This must be specified. It would be a good idea to measure both envelope and skeletal densities and then calculate the porosity of your material.

*We now describe that we measured the bulk density (i.e. the envelope density) by dividing weight of the sample by the volume (surface area * thickness) of the material (l. 178-179). We have not been able to determine the skeletal densities due to lack of equipment.*

2) Figure 5. is definitely not adapted. It is reproduced. Adapted would indicate that you have changed it significantly. Adding your data to it does not suffice as an adaptation. You will need to ensure that you have copyright permission for the use of the image.

We changed the legend and now describe that we had permission of Granta design (see legend Figure 4).

3) The water contact angle can be prone to very significant error. Have you used a static or dynamic contact angle measurement technique? A static water contact angle may be subject to significant contact angle hysteresis and will likely be inaccurate. Please provide advancing or receding dynamic contact angle measurements. Preferably both. 3 replicate drop profiles is also insufficient for contact angle measurements. Please provide at least 10 replicates.

We used static water contact angle measurements (l. 100, 131, 185, 200, 257) and increased the number of measurements to 15 technical replicates for each of the 4 biological replicates. Data show low variation and are statistically different between sample types.

4) The contact angle measurements do show changes in hydrophilicity. However, water permeability is indicated by water flux. I do not see any water flux measurements in this article. These must be included in order to claim differences in water permeability.

We have removed our claim about water permeability.

5) You say: “The intra- and extracellular glycerol would also explain why mycelium treated with 8–32% glycerol did not sorb water, while sorption was observed in the case of untreated mycelium despite its highly hydrophobic nature of its surface”. However, you have not measured sorption at all. Contact angle measurements only provide you with the contact angle and if you use more than one test fluid then an approximation of the surface energy. You cannot talk about sorption when you haven’t measured sorption.

We have now performed the water absorption experiment and simultaneously measured thickness expansion (l. 105-113). Results confirmed our initial claim.

6) You say: “Scanning electron microscopy indicates that glycerol fills the air voids that are present in untreated mycelium”. These SEM micrographs should be included in the article or supplementary material.

We have changed the statement to “The number of air voids in untreated mycelium decreased after treatment with 8% glycerol” as is shown in Figure 1 (l. 73-74).

7) You say: “For instance, the reduced water sorption of mycelium materials after treatment with glycerol is of interest for outdoor applications.” However, this is contradictory to your results. The more glycerol you add the more hydrophilic your materials become (reducing contact angle). So, treatment with glycerol certainly isn’t in the interest of outdoor applications. Also, if you are going to talk about water sorption then it should be accompanied by an isotherm. From what I can see you haven’t measured anything at all related to water sorption.

We have now included data about water absorption of our fungal material (see point 5 of Reviewer 1) and removed our claim regarding outdoor applications.

8) This article addresses the mechanical and surface properties of treated mycelium biomass. You should compare your results to other studies examining the mechanical and surface properties of treated mycelium biomass. e.g. Jones, M., Weiland, K., Kujundzic, M., Theiner, J., Kählig, H., Kontturi, E., John, S., Bismarck, A. and Mautner, A., 2019. Waste-derived low-cost mycelium nanopapers with tunable mechanical and surface properties. *Biomacromolecules*, 20(9), pp.3513-3523.

We have now included these data (l. 37-43).

9) The reference Jones, Huynh, Dekiwadia, Daver, & John, 2017 appears in text but does not appear in the reference list.

We have now included the reference (l. 34).

Reviewer #2

This manuscript reports a sustainable fungal mycelium, the production process and experimental measure methods were present, and the test results were analyzed and discussed. Finally, the produced fungal mycelium was figured on the material family chart, and shown the mechanics behaviors compared with other materials. This work is interesting for the sustainable green

materials, and I think it could meet our journal. However, this manuscript should clear some mechanical test process and method in detail. The comments please find as follows:

1) Line 87-88: How much the light intensity

We have added the light intensity of 1000 lux (l. 155-157).

2) Line 102: What size of the samples for mechanical tests?

We have added the dimensions of the specimens (Figure 5; l. 172-174).

3) Line 105: During the preparation of tensile test, please clear that how to fix the top and bottom of the size?

Specimens were clamped 10 mm from each end of the sample (l. 177-179).

4) Line 106: What is the temperature during the test?

Tests were performed at room temperature (l. 175-177).

5) Line 107 What experiment standard was used to test the Young's modulus? Generally, Young's modulus could be determined from the compressive test.

We have used tensile tests to determine the Young's moduli of materials (l. 175-177).

6) Line 108-109: the symbol of ultimate tensile strength should be the σ , and the strain should be ϵ .

We have changed the symbols (l.180-182).

7) Line 101-109: How many samples were measured for each group test.

The number of samples of all experiments have now been added (Table 1, l. 66-67, 186-187).

8) Line 108: How to determine the ultimate tensile strength from the stress-strain curves (For the figure 3)? Generally, the peak of stress-strain curve or the stress at certain strain is determined as the tensile strength. So, please clear the determine method and test standard in detail.

The ultimate tensile strength was determined by the maximum load per unit area of the specimen (l.180-182).

9) Line 146 Figure 2: Please show SEM photos to present the micro-structure of the mycelium.

We have added SEM images with increased resolution (Figure 1). However, it is still hard to discriminate individual hyphae. This has now been described (l. 70-72).

10) Line 180 Figure 4C, Please replace the vertical coordination axis as 'broken strain'.

We have changed the title of the y-axis to strain at failure and now show this Figure as Supplementary Figure 1 (see point 20 of Reviewer 3).

Finally, I encourage the authors to show the thermal conductivity and toxicity of these fungal mycelium.

We will include thermal conductivity once we are investigating specific applications of our materials. Note that Schizophyllum commune is an edible mushroom (l. 45-47) and not reported to produce mycotoxins.

Reviewer #3

The study showcases that a treatment with glycerol impact mycelium-material properties, resulting in sheets with stiffer and more elastic properties, similar to industrial polymers and elastomers. This study shifts mycelium materials to propose relevant alternatives to synthetic raw materials, an important milestone in promoting this research field. The methods and results are properly detailed and clearly described. The obtained results provide useful information to develop further applications and production methods. The use of pure mycelium material originated in liquid culture instead of producing plant-based mycelium composites is novel and promising, yet to my knowledge it should not be targeted as the novelty of this article. I suggest adding some references to other publications that did similar studies using mycelium material originated in liquid culture (detailed below) or refer to other bio-based materials that offer similar properties – it will contribute to emphasize the uniqueness of your approach and locate your results within a focused context.

We have reformulated the part on the growth conditions and material production (l.58-60) and refer to Jones et al. (2019) that also used liquid shaken cultures (l.33-35).

1) Lines 21-22: This sentence could be refined since mycelium materials in academia and industry are regularly produced in liquid cultures. Please consider referring to Livne, A. et al. (2019) – To my knowledge they demonstrated the use of shaken liquid culture in a recent publication [A fungal mycelium templates the growth of aragonite needles. J. Mater. Chem. B 7, 5725–5731].

Indeed, fungal mycelium is routinely produced as liquid shaken cultures in industry and academia, however, not in the scope of producing bio-based materials. Livne et al., 2019 produced liquid grown mycelium to precipitate calcium carbonate. Jones et al. (2019) did grow liquid shaken cultures and is now being cited (see introducing point Reviewer 3).

2) Lines 21-22: Maybe replace ‘material’ with ‘films’

We have replaced ‘materials’ with ‘films’ (l.20).

3) Try to avoid clustered references, maybe separate the mycelium-based references from the other sourced materials

The grouped references are now separated based on mycelium references and other sourced materials (l.33-35).

4) What are the expected advantages of mycelium-based materials over the other bio-based materials that you mention?

Advantages have now been added to the introduction (l.35-36).

5) Lines 50-66: I don’t think that this paragraph and the following one are essential – It would be better to give some background on the use of glycerol as a plasticizing agent, why glycerol, its

effect on physical and mechanical properties, is it biodegradable? What other relevant properties does it offer, maybe a short overview on previous starch/chitosan/glycerol films studies

We have removed these paragraphs and introduced glycerol (l. 52-57).

6) What are the relevant differences between using chitin, chitosan and mycelium?

We have introduced glucan/chitin/chitosan materials and describe the difference with mycelium (l.37-51).

7) Line 57, line 69: Here again I believe it would be clearer to separate the references, for example to pure and composite materials.

Line 57 has been removed. References are now separated (l.44-45).

8) Line 78: Worth regarding to W. Nawawi et al. 2019 that recently produced mycelium materials originated in liquid culture, I think it is comparable and relevant, and maybe suggest what considerations led you to use mycelium rather than extracted chitin?

We have now included the paper of Nawawi et al 2019 (l. 37-43). However, the materials in the paper originate from mushrooms and not from liquid cultures. We have used pure mycelium without any further processing because this simplifies the production process.

9) Line 87: Do you think that the Minimal Medium might have contributed to the increase in strength/or density?

It is very unlikely that the minimal medium (MM) contributes to the mechanical strength of the material. The inorganic constituents of MM are filtered out during the harvesting process and only small amounts may have attached to the mycelium.

10) Line 88: "The culture was homogenized in 100 mL MM for 30 sec" At what rate?

We have added the rate at which the mycelium was homogenized (l.157-159).

11) Line 90: What does the minimal medium contain in this stage? Is it in a liquid state here? Does this substrate contain glucose? What else?

The composition of MM is described in Reference 17 and contains glucose as carbon source and asparagine as nitrogen source, both now being added in the Methods (l.155-157). The pre-culture and culture are both grown as liquid shaken cultures (l. 157-161).

12) Line 91: "grown for 7 days in the dark at 200 rpm" At what temperature?

We have added the incubation temperature of 30 °C (l. 161).

13) Line 93: Was the mycelium deactivated? Add drying conditions (temperature, humidity, clamping?)

The mycelium was not heat killed but dried at room temperature at a RH of ±50% while being covered by cellophane (l. 163-165).

14) Line 135 (figure 1): Is it a light microscope or camera? Do you discuss the effect of inoculum volume on pellet dimensions in the text? Are there other factors that might affect it? Why is the pellet dimension important? Does it affect other properties or affected by certain factors during growth?

We reconsidered the addition of the pellet size and concluded that it is not interesting enough for this paper and removed it from the manuscript.

15) Line 139: Replace 'incubated' with 'submerged'?

Replaced as suggested (l. 70).

16) Line 145 (figure 2): The scale bars are different thicknesses and should be fixed

Corrected as suggested (now Figure 1).

17) Maybe add a top view photo to better understand what your material looks like

The top view does not really add additional information. We did add that A and D refer to longitudinal sections of the mycelium films (legend Figure 1).

18) Line 155 (Table 1): mm units would be more appropriate for film thickness

Adapted as suggested (Table 1).

19) I wonder why all the standard deviations for thickness are zero (Table 2). Did you take only 1 measurement? you can compare your measurements to the microscope images to add data and validate your measurements.

We wanted to have the same number of decimals throughout Table 1 (former Table 2). Since the thickness is now shown in mm (see comment 18 of Reviewer 3) also the SEMs are now visible.

20) Data in Table 1 is also presented in figure 3 and 4. Maybe the table could go to supplemental data or that the figures should present something beyond the data of the table.

Figure 2 (former Figure 3) shows the stress / strain curves not visible in Table 1. Indeed, data of the original Figure 4 was a repetition of data in Table 1. Since Table 1 shows an overview of all properties we have decided to relabel the original Figure 4 as Supplementary Figure 1.

21) Line 167: Colors should be more varied – maybe add shapes or letters near each curve?

We have adapted the Figures following the instructions of the journal (now using lettering).

22) Line 170: Replace 'incubation' with 'treatment' or 'submerged'?

We have changed "incubation" into "submersion of mycelial films" (l.87-90).

23) Line 180 (figure 4): This information already appears in table 1 - Can you show new data here? For example, curves that represent interesting crossing of different properties.

See point 20) of Reviewer 3.

24) Line 189: Contact angle is affected by the surface roughness- do you know whether the roughness of the films was affected by the different compositions you tested? And glycerol is hydrophilic but maybe it also contributed to surface smoothness thus decreased its hydrophobicity?

We have implemented these comments in the Discussion (l.136-137).

25) Line 191: Not clear if the water were taken-in faster or slower with glycerol. I think you should refine if you are saying that the water droplets did not penetrate the films after 5 min at 32% glycerol.

We have adapted this sentence and added data on water submersion (l.105-113).

26) Line 195-208: Most of the paragraph discusses the previous study. I would recommend to significantly reduce it or focus on why it is relevant in the discussion

We have condensed the beginning of the discussion by 50% (l. 116-123).

27) Line 205: Then what is the difference between static and liquid culture? The yield? Based on your studies- which one suits better for industrial production of mycelium materials?

We now describe the pros of liquid shaken cultures (l.118-120).

28) Line 207: Polymer-like...

This has now been corrected (l.120-122).

29) Line 208: "Thus, we have produced for the first-time mycelium materials that are classified in the latter material group" > I think that this is the important point to emphasize for 'first time' and try to avoid erosion of this phrase over this article.

We have rephrased this sentence (l.122-123).

30) Line 195-208: "16 or 32% glycerol is higher than that of the plasticizing agent" > Confusing- If I got it right, the glycerol is the plasticizing agent. Please refine if you are saying that the density of the mycelium-glycerol is greater than pure glycerol?

We have removed this sentence.

31) Line 227: I agree that water is probably trapped between the hyphae, but what do you mean with the phrase 'hydrophobic air'? Did you mean just air?

We have rephrased the paragraph and removed our statement about hydrophobic air.

REVIEWERS' COMMENTS:

Reviewer #1 (Remarks to the Author):

My concerns have been largely addressed. So all good regarding my assessment of this submission.

Reviewer #2 (Remarks to the Author):

Authors addressed the comments points by points, this current version is acceptable one.

Specially comments:

Figure 2 and Line 182-183: Authors should clear that the last test point on the stress-strain curve is the maximum value in the test.

Reviewer #3 (Remarks to the Author):

The manuscript presents an original study with detailed materials and methods that contribute and promote the academic dialog around the emerging field of mycelium-based materials.

Attached are a few little comments - No need for further review

Line 126-130: What step caused the washing of water-soluble proteins and sugars?

Does the mycelium neutralize the hydrophilicity of glycerol?

what is assumed to be the uptake/affiliation mechanism between mycelium and glycerol (structural and chemical).

line 151: Might be worth adding what kind of applicable directions does the post treatment with glycerol adds compared to pure mycelium sheets

Lines 162-3: Isn't there an additional homogenization step before filtration? And for reproducibility maybe add what was the %dry weight of the mixture before it was processed into films.

Point-by-point response to the referees' and editor comments

Again, we would like to thank the editor and reviewers for helping us to improve the quality of the manuscript. We have addressed the AIP checklist table (see submitted file) and all final points of the reviewers. Our response is in italics and we refer to line numbers (abbreviated with l.) in the revised manuscript.

Reviewer #1 (Remarks to the Author):

My concerns have been largely addressed. So all good regarding my assessment of this submission.

Reviewer #2 (Remarks to the Author):

Authors addressed the comments points by points, this current version is acceptable one. Specially comments: 1) Figure 2 and Line 182-183: Authors should clear that the last test point on the stress-strain curve is the maximum value in the test.

For most of the measurements this was indeed the case. However, for treatment of films with 16 % and 32 % glycerol, the last test point was somewhat lower than the maximum value (Fig 2 g,h). In these cases, the software continued to record the stress for a very short time after the sample broke. Hence, it recorded lower stress values. Therefore, we have addressed the ultimate tensile strength being obtained using the maximum load recorded (l.181-183).

Reviewer #3 (Remarks to the Author):

The manuscript presents an original study with detailed materials and methods that contribute and promote the academic dialog around the emerging field of mycelium-based materials. Attached are a few little comments - No need for further review

Line 126-130: What step caused the washing of water-soluble proteins and sugars? Does the mycelium neutralize the hydrophilicity of glycerol? what is assumed to be the uptake/affiliation mechanism between mycelium and glycerol (structural and chemical).

We have now addressed the cause of washing out water-soluble proteins and sugars (l.123-124) and the assumed uptake/affiliation mechanism between mycelium and glycerol (l.126-130).

line 151: Might be worth adding what kind of applicable directions does the post treatment with glycerol adds compared to pure mycelium sheets

We have now related the properties of our materials to materials that are in the market at the moment; thus giving a better idea of possible applications (l. 149-152).

Lines 162-3: Isn't there an additional homogenization step before filtration? And for reproducibility maybe add what was the %dry weight of the mixture before it was processed into films.

There was no additional homogenization step before filtration; intact pellets were used to produce the films (l. 54-55). The % dry weight has now been added in the Result section (l. 63-64).